∂ | **Open Peer Review** | Bacteriology | Research Article

# The whole-genome molecular epidemiology of sequential isolates of *Acinetobacter baumannii* colonizing the rectum of patients in an adult intensive care unit of a tertiary hospital

Dieter Bulach,[1,2] Glen P. Carter,[3] Lucy Li,[3] Ghayda Al-Hashem,[4] Vincent O. Rotimi,[4] M. John Albert[4]

**ABSTRACT** *Acinetobacter baumannii* is a nosocomial multidrug-resistant pathogen that specifically colonizes and infects patients in intensive care units (ICUs). Genome plasticity of *A. baumannii* may contribute to the rapid development of *A. baumannii* resistance during patient treatment. There is little clarity on the dynamics of colonization of *A. baumannii* in humans. We studied 269 serial isolates of *A. baumannii* colonizing the rectum of 32 adult patients in an ICU; 5 to 16 isolates were obtained from each patient over a period of weeks to months. The isolates were sequenced using the Illumina platform. Multi-locus sequence typing revealed genetic diversity with 13 sequence types (STs); the dominant ST2 accounted for 202 of the 269 isolates and was isolated from each patient. A core genome comparison of isolates was used to elucidate a detailed understanding of the relationship among isolates within an ST and the genomic relationship among STs. One to three STs were observed in the isolates from each patient. Instances of unchanged colonization, sequential lineage colonization, and colonization with multiple lineages of *A. baumannii* were seen. The number of resistance genes carried by the isolates varied from 2 to 17. The predominant genes corresponded to those encoding resistance to β-lactam and aminoglycoside antibiotics. Generally, there was a strong correlation between ST and a particular antimicrobial resistance gene profile. This study makes an important baseline contribution to information that will help us understand the role of *A. baumannii* colonization in the development of opportunistic infections in ICU patients.

**IMPORTANCE** *Acinetobacter baumannii* is a multidrug-resistant nosocomial pathogen that colonizes and infects debilitated patients in the ICU. There is very little information on the genomic characteristics of colonizing strains. This information is important to understand the evolution of lineages of *A. baumannii* that develop resistance while patients receive antibiotic treatment in the ICU. Our study demonstrated different patterns of colonization of the rectum of ICU patients with different STs of *A. baumannii* while one ST colonized all patients. Some STs carried more antibiotic resistance genes compared to others. However, there was a correlation between ST and a particular resistance gene profile. Our results further elucidate the dynamics of enteric colonization of this opportunistic pathogen.

**KEYWORDS** *Acinetobacter baumannii*, whole-genome sequencing, molecular epidemiology, rectal colonization, intensive care unit

A cinetobacter baumannii is an aerobic, Gram-negative, catalase-positive, oxidase-negative, non-motile, and non-fermenting coccobacillus (1). It is found in wet environments and food items, including meat and vegetables, and is part of the microbial flora of many animals (2–4). It has attained notoriety as a nosocomial, opportunistic

Address correspondence to M. John Albert, manuel.albert@ku.edu.kw.

The authors declare no conflict of interest.

pathogen, causing problems in intensive care units (ICUs) affecting debilitated patients with compromised immunity, disrupted normal microbial flora, or barrier immunity (5). *A. baumannii* causes a variety of infections, including skin and soft tissue infections, ventilator-associated pneumonia, urinary tract infection, blood-stream infection, wound infection, and meningitis (6–9). The persistence of *A. baumannii* in hospitals outside the human host, and in particular on ICU equipment, poses a significant risk to critically ill patients as they are admitted to ICU (10). The combination of environmental persistence in hospitals and the relatively high frequency of *A. baumannii* carriage means that in hospitals, both endemic and outbreaks of *A. baumannii* are usually polyclonal (11–13). Environmental persistence of *A. baumannii* in combination with the aggressive use of antibiotics to treat critically ill patients in ICU is likely to contribute to the selection of multi-resistant lineages of *A. baumannii*. The ICU in Mubarak Al Kabeer Hospital, Kuwait has witnessed several *A. baumannii* outbreaks with some confirmed as polyclonal (14). In addition, other hospitals in Kuwait have reported cases of *A. baumannii* infection (15, 16). The management of nosocomial infections caused by *A. baumannii* is an ongoing concern in Kuwait. The current study used genomics to investigate changes in *A. baumannii* colonization after ICU admission. Using a genomic analysis, evaluation of lineage persistence/exchange and determinant acquisition (particularly antimicrobial resistance genes) by *A. baumannii* during colonization was carried out. This investigation enabled us to determine if internationally significant lineages of multi-resistant *A. baumannii* were present in Kuwait. This study expands our understanding of the epidemiology of this important opportunistic pathogen.

## MATERIALS AND METHODS

### Patients and bacteria

The same cohort of patients used in the previous two studies has been used in the current study. The relevant details have already been published (17, 18). Briefly, the study consisted of 32 adult patients admitted to the ICU of Mubarak Al Kabeer Hospital, Jabriya, Kuwait, all of whom were identified as having gastrointestinal tracts colonized long term with *A. baumannii*. The median age of patients was 65 years, with an interquartile range of 50.25–69.75 years, and the cohort was gender balanced. For this study, long-term colonization with *A. baumannii* was defined as having positive cultures on ≥5 consecutive rectal swabs post-admission to the hospital. Swabs were collected on the day of admission, the third day after admission, and then twice weekly until the patient was discharged or dead. The duration of specimen collection was from 14 to 343 days, the number of times a patient was sampled was from 5 to 11, and the number of isolates studied per patient ranged from 5 to 16. The details of the sampling regime are given in Table S1. Specimens were collected between March 2015 and June 2016. Isolation of *A. baumannii* from swabs was performed by enrichment culture and growth on selective agar. Isolates were confirmed using API 20 NE and an *A. baumannii*-specific multiplex PCR assay based on *gyrB* gene (19, 20).

### Whole-genome sequencing

Genomic DNA from *A. baumannii* isolates was extracted using the DNeasy blood and tissue extraction kit (Qiagen), sequencing libraries were prepared using the Nextera XT DNA sample preparation kit (Illumina), and the sequence read data were produced on the Illumina NextSeq instrument (paired end, 150 base reads).

### Genome assembly

A genome sequence was produced from the read data using Spades v3.9 (21). An isolate purity check and confirmation of taxonomic classification were performed using kraken2 (https://github.com/DerrickWood/kraken2) and the GTDB kraken2 database (https://gtdb.ecogenomic.org/ produced using https://github.com/leylabmpi/Struo2).

## Genome comparison

Core genome comparison of isolates was performed using Snippy (https://github.com/tseemann/snippy) and IQtree (https://github.com/iqtree/iqtree2) as implemented in Bohra (https://github.com/MDU-PHL/bohra). The reference genome used for core genome comparison was from ACICU strain (accession no. GCF_000018445.1). The reference-free K-mer-based genome comparison was performed using Mashtree (https://github.com/lskatz/mashtree).

## Gene detection and sequence typing

The multi-locus sequence type (MLST) of each isolate was determined from the assembled genome sequence using the MLST software (https://github.com/tseemann/mlst) and the *A. baumannii* MLST scheme developed at Institut Pasteur (http://pubmlst.org/abaumannii). The antimicrobial resistance gene content for each isolate was determined from the assembled genome sequence using amrfinder plus (https://github.com/ncbi/amr) and the NCBI AMR reference gene catalog (https://www.ncbi.nlm.nih.gov/pathogens/refgene/).

## Patient longitudinal colonization profiles

Sequential isolates were used to classify colonization to a particular profile. Expected profiles were (i) no change in sequence or minor changes due to single-nucleotide polymorphisms (SNPs) and indels indicating persistence of the infecting strain, (ii) evidence of mixed infection with different strains being isolated alternately or concurrently, and/or (iii) disappearance of the original infecting strain followed by infection with a different strain.

## RESULTS

### *A. baumannii* isolates

A total of 493 patients were screened during the study period. We selected patients from whom *A. baumannii* was isolated on ≥5 consecutive occasions. Accordingly, 270 isolates from 32 patients were included in the study; patients were identified in the study by letters (A to Z, AA, AB, AC, AD, AE, and AF). The isolates were named by patient identifier, chronologically ordered sample number (per patient), and colony designation (where multiple colony morphotypes were sampled). For example, isolates C5a, C5b, and C5c are from patient C at the fifth sampling point with three different colony morphotypes, a, b, and c. One of the 270 isolates, namely isolate E5, could not be revived for sequencing and was thus excluded.

### Genome length, contig, and sequence type

Isolate sequencing information, genome assembly statistics, and sequence typing are shown in Table S2. Evaluation of read sets using kraken2 confirmed the laboratory taxonomic classification of isolates and the purity of isolates. The 269 isolates assembled with an average genome size of 3.8 Mb (range: 3.5 Mb to 4.2 Mb), and the number of contigs ranged from 50 to 772. Among the 269 isolates, there were 13 different STs, including 3 new STs—ST2334, ST2336, and ST2337. The most frequently identified sequence types were ST2 (202 isolates), ST113 (24 isolates), and ST584 (13 isolates); each of the remaining STs was identified in fewer than 10 isolates.

### Nucleotide differences in the core genomes

When a core genome comparison of all 269 isolates was performed using the genome sequence of *A. baumannii*, ACICU strain as the reference genome sequence (accession: GCF_000018445.1), groups of isolates determined by ST occurred as monophyletic clades in the inferred tree. A visual overview of the relationship among isolates is shown

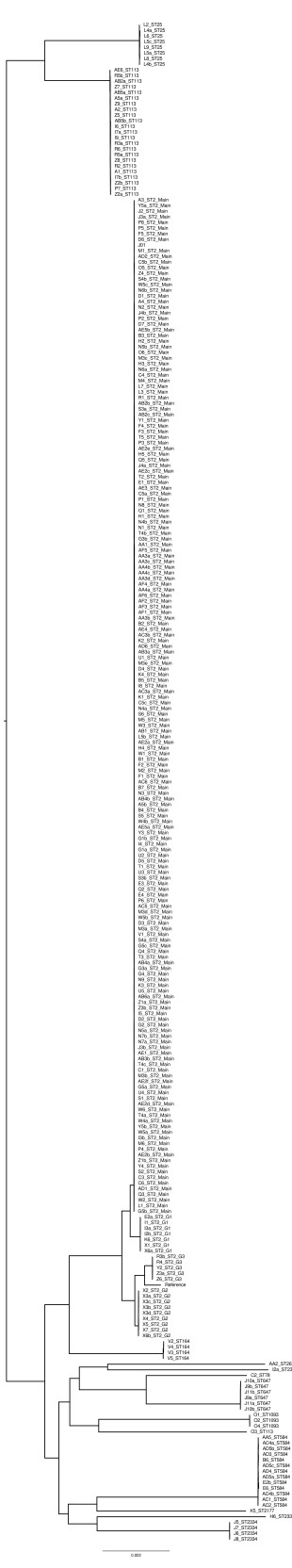

**FIG 1** A phylogenetic tree showing the inferred core genome relationship for the 269 *A. baumannii* isolates from 32 ICU patients in Kuwait. The sequence type of each isolate is shown on the taxon label. The reference genome used in the analysis was *A. baumannii* strain ACICU (accession GCF_000018445.1). (Continued on next page)

**FIG 1** (Continued)

The core genome includes ~77% of the reference genome sequence. The most distantly related isolates are isolates I2a and H6 with 33,878 pairwise SNP differences.

in Fig. 1. Each ST detected in the study represents a distinct lineage of *A. baumannii* (pairwise SNP distance to most related isolate outside the ST is ≥15,000). Notably, isolates within an ST in this study are highly related (pairwise SNP distance within an ST is <40). The exception is ST2. In the core genome comparison, the clade that included all 202 ST2 isolates comprised a group of 181 isolates that had a maximum pairwise SNP distance of 40. The remaining 21 isolates were grouped in three small clusters, G1 (E2a, I1, I3a, I2b, K6, X1, and X6a; intragroup: maximum pairwise SNP distance = 20; pairwise distance to main ST2 group = 1,490), G2 (X2, X3a, X3c, X3b, X3d, X4, X7, X5, and X6b; intragroup: maximum pairwise SNP distance = 20; pairwise distance to main ST2 group = 2,700), and G3 (R3b, R4, Z3a, Y2, and Z6; intragroup: maximum pairwise SNP distance = 10; pairwise distance to main ST2 group = 4,400). To investigate the relationship of the Kuwait ST2 isolates with the international ST2 isolates, the genomic relationship between the Kuwait ST2 isolates and a set of 288 ST isolates with closed genomes (Table S3) was investigated using Mashtree and summarized as a phylogenetic tree in Fig. S1. The four clusters of Kuwaiti ST2 isolates appeared as clusters in Fig. S1 (colored with green [main], red [G1], dark blue [G2], and light blue [G3] branches in the tree). The main group of 181 Kuwaiti ST2 isolates is in a cluster that is interspersed with 15 isolates predominantly from the USA, China, and Belgium.

Table 1 shows the distribution of STs detected in each patient. In 12 out of 32 patients, the same lineage (i.e., ST) was detected at each isolation. In all cases, these were ST2 isolates. This did not include one patient (patient X) for whom the ST2 sub-lineage differed in 2 out of 11 isolates. Among the remaining 19 patients, isolates from two or three STs were detected. Remarkably, ST2 was detected in all patients. The frequency distribution of STs and changes of STs over time for each patient are shown in Table S1. In some patients, the same STs persisted; in others, there were mixed infections during certain sampling points; and yet in others, the initial STs were replaced by other STs.

## Antimicrobial resistance genes

An overview of the antimicrobial resistance genes found in the 269 isolates is shown in Table S4. Genes belonged to seven different classes, the prominent ones being those conferring resistance to β-lactam and aminoglycosides. A comparison of phenotypic resistance and antimicrobial gene resistance is shown in Table S5. Within lineages, in this instance defined by ST and subgroup (for ST2), isolates have consistent patterns of resistance gene content and detected resistance phenotypes. Although there are some clear exceptions, W6 is an ST2 (main subgroup) isolate with a typical ST2 isolate gene profile, but this isolate is sensitive to all tested antibiotics. In addition, B6 is resistant to many antibiotics despite other ST584 isolates having the same gene profile and limited phenotypic resistance. The ST584 isolates differ from other isolates in the study, in that they are TET resistant with no *tet* resistance genes detected. However, across all isolates, there was a good correlation between detected point mutations in the *gyrA* and *parC* genes and a fluroquinolone-resistant phenotype.

A comparison of ST and antimicrobial resistance gene profile of the 269 *A. baumannii* isolates is shown in Table S6. Notably, ST2 (10–14 genes) and ST25 (10–17 genes) isolates carried more antimicrobial resistance genes than other STs. ST113 and ST164 isolates carried an intermediate number of antimicrobial resistance genes (seven to nine genes), and isolates from the remaining STs carried fewer than four antimicrobial resistance genes. Within STs, there were usually groups of isolates with the same antimicrobial resistance gene profile (ST78, ST267, ST584, ST647, ST1093, ST2177, ST2334, ST2336, ST2337). However, some STs had multiple resistance gene profiles—14 for ST2, four for ST113, and two each for ST25 and ST164.

**TABLE 1** Sequence types of *A. baumannii* isolates obtained from each patient

| Isolates from patient | No. of STs | ST(s) |
|---|---|---|
| A | 2 | 2, 113 |
| B | 2 | 2, 584 |
| C | 2 | 2, 78 |
| D | 1 | 2 |
| E | 2 | 2, 584 |
| F | 1 | 2 |
| G | 1 | 2 |
| H | 2 | 2, 2,336[a] |
| I | 3 | 2, 113, 2,337[a] |
| J | 3 | 2, 647, 2,334[a] |
| K | 2 | 2, 2,177 |
| L | 2 | 2, 25 |
| M | 1 | 2 |
| N | 1 | 2 |
| O | 2 | 2, 1,093 |
| P | 2 | 2, 113 |
| Q | 1 | 2 |
| R | 2 | 2, 113 |
| S | 1 | 2 |
| T | 1 | 2 |
| U | 1 | 2 |
| V | 2 | 2, 164 |
| W | 1 | 2 |
| X | 1 | 2 |
| Y | 1 | 2 |
| Z | 2 | 2, 113 |
| AA | 3 | 2, 267, 584 |
| AB | 2 | 2, 113 |
| AC | 2 | 2, 584 |
| AD | 2 | 2, 584 |
| AE | 2 | 2, 113 |
| AF | 1 | 2 |

[a]New STs found in the study.

The comparison of bacterial colonies with STs and antimicrobial resistance gene profiles is shown for 21 patients (A, C, E, G, I, J, L, M, N, R, S, T, W, X, Y, Z, AA, AB, AC, AD, AE) who had multiple colonies tested at different sampling times. In 12 instances, colony differences showed genotypic differences; but in 34 instances, colony differences did not result in genotypic differences (Table S7).

The antimicrobial resistance gene profiles of sequential isolates in patients who were colonized with a single ST and the same clade group were examined. Isolates from ten patients (D, F, G, M, N, Q, S, T, U, AF) were qualified for this analysis. The isolates from all patients were largely stable in their resistance gene content except for a few isolates which showed either some increase or decrease in their gene content (Table S8).

## Dynamics of colonization

The analysis of the isolates from 32 patients showed that 10 patients were colonized with the same ST for the duration of their stay in ICU. A further 17 patients were sequentially colonized with two or three STs for the duration of their stay in ICU. The five remaining patients were colonized with multiple STs at various times during their stay in ICU. Given that single colony sampling was predominantly used, this may have led to an underestimate of colonization with multiple STs.

## DISCUSSION

The genome characteristics of our *A. baumannii* isolates—read set, number of contigs, and genome size—were like those published previously using the Illumina sequencing technology (22–25).

Some studies have indicated that genome changes occur in *A. baumannii* isolates during antibiotic treatment (26, 27), during outbreaks (28), during studies in a hospital setting over a period of days (29, 30), or on sampling of sites other than the digestive tract of a limited number of patients (31–33). It was found that genome changes due to mutation, recombination, and content can occur in a short period of days. Homologous recombination is associated with loss or swapping of genes encoding surface molecules. Variation in surface molecules is mediated by horizontal gene transfer. Genetic variation is also due to mobile elements, such as insertion sequences and plasmids, mobilizing parts of the chromosome. The rate of mutation is estimated to be −5 to −10 per year. However, in some instances, a higher rate of −24 has been observed. Mutations can be synonymous as well as non-synonymous. Mutated strains can retain the same STs. Reinfections and co-infections of the patients by similar but different strains can also occur.

Host colonization with *A. baumannii* is likely to play an important role in the development of infections that lead to disease caused by this opportunistic pathogen. Intra-lineage and inter-lineage diversity of the *A. baumannii* colonizing the host prior to disease may contribute to the apparently rapid adaption observed during antimicrobial treatment. This study investigates longitudinal changes in the host colonization of 32 patients while in ICU.

The predominant sequence type of *A. baumannii* in our study was ST2. ST2 belongs to international clone II (ICII) (34). *A. baumannii* ICII is associated with nosocomial outbreaks and multidrug resistance (35) and has spread globally (36). *A. baumannii* ST2 isolates frequently carry multiple antimicrobial resistance genes, which were also observed among the Kuwaiti ST2 isolates. Previous studies have proposed that persistence in the hospital environment may play a role in the global distribution of this ST. We observed that all patients were either colonized by ST2 at the start of their ICU stay or became colonized during their stay. The strong ability of ST2 isolates to colonize the human gastrointestinal tract and a general increase in the mobility of humans will also contribute to global distribution. The *A. baumannii* isolates from other hospitals in Kuwait should be studied to ascertain whether ST2 is the predominant type in these hospitals. Like ST2, there are other STs that appear to be multidrug resistant based on their antimicrobial resistance gene profile, and these STs contrast with STs that carry very few antimicrobial resistance genes. The coexistence of these "sensitive" ST lineages with more resistant lineages suggests there are some barriers to horizontal movement of antimicrobial resistance genes in *A. baumannii*. It would be interesting to track the epidemiology of these more "sensitive" STs; some of which may not be distributed widely. We note that as part of this study, we have identified three new STs that have not been observed outside Kuwait. While this longitudinal study was performed over a relatively short period of time and involved a small number of patients, we have been able to observe genomically stable lineages. By extending this use of isolate genomics to isolates from a broader cross-section of the population (including healthy individuals), we are likely to develop a lineage/ST-level understanding of the epidemiology of this important opportunistic pathogen.

The *A. baumannii* isolates in our study were multidrug resistant and carried several antimicrobial resistance genes. The diseases for which the patients were admitted in the ICU were chronic respiratory disease, diabetes mellitus, cerebral vascular accident, chronic kidney disease, coronary artery disease, etc. These patients received multiple antibiotics during the stay in the ICU. The most common antibiotics received were ceftriaxone, meropenem, piperacillin/tazobactam, and colistin. Seven patients (B, D, F, J, Q, Y, and AF) had only rectal colonization with no clinically proven infection with *A. baumannii*. The remaining 25 patients had infection with *A. baumannii* in

addition to colonization of the rectum. These 25 patients were given colistin to treat the *A. baumannii* infection. The antibiotics to which the isolates were tested were piperacillin/tazobactam, ceftazidime, cefepime, imipenem, meropenem, colistin, gentamicin, amikacin, tetracycline, tigecycline, ciprofloxacin, levofloxacin, and trimethoprim-sulfamethoxazole. The prevalence of resistance to all antibiotics except colistin and trimethoprim-sulfamethoxazole varied from 73.7% to 95.9% to colistin was 2.9% and to trimethoprim-sulfamethoxazole was 31.1%. However, no correlation was observed between the phenotypic resistance of the isolates and the administered antibiotics for treatment (18). While we have observed some SNP-level changes in sequential isolates (as evident in the construction of the phylogenetic tree in Fig. 1), there is no clear indication of in-host selection of resistance. However, the clear trend was for the selection of different lineages in the host either from re-infection or the carriage of multiple lineages. There was largely a correlation between the presence of resistance genes and the corresponding phenotypic resistance in our isolates. In cases where a correlation could not be established, might be attributed to regulatory genes and efflux pumps.

The two common patterns were colonization with two or three different lineages/STs sequentially or with the same lineage/ST throughout the period of the study (see Table 2). More complex patterns of colonization were observed in five patients and could not be resolved with the sampling strategy used in the study. In these patients, multiple lineages/STs were isolated from one sample, thus indicating co-colonization with multiple lineages. There was an unequal sampling frequency of patients (Table S1). How this has impacted the strain distribution is not clear. Also, given that single colony sampling was predominantly used, this may have led to an underestimate of colonization with multiple STs. The number of colonies selected from each sample was based on the different colony morphotypes present. This was based on our previous observation in specimens from 12 patients where a correlation was observed between the difference in colony morphology and the difference in genotype that was based on the DeversiLab typing system (17). However, our observation in the current study suggested that there is a poor correlation between morphotype and genotype that is based on ST and the resistance gene content.

Using the same cohort of patients, we previously investigated the antimicrobial susceptibilities of 270 serial isolates from all 32 patients to investigate whether the gut of the patients was the site of origin of polyclonality of *A. baumannii* infection. It was found that the isolates were multidrug resistant, and there was no consistent pattern of resistance in serial isolates from the patients. The isolates from the patients appeared to be polyclonal by their resistance patterns, but no firm conclusion on the origin of polyclonality could be drawn (18). In another study using the same isolates, we attempted to define the genetic relationship of 180 selected serial isolates from 13 patients. The isolates were analyzed by pulsed-field gel electrophoresis, ST, and clonal complex by eBURST analysis. It was found that serial isolates constituted a mixture of identical, related, and unrelated pulsotypes. These data suggested a combination of an initial colonizing isolate undergoing mutation and colonization by independent isolates.

**TABLE 2** Summary of colonization details of patients with sequential *A. baumannii* isolates

| Colonization details | No. of patient(s) | Patient ID(s) |
|---|---|---|
| Colonized with two STs, sequentially | 11 | B, C, H, P, V, X, Y, AB, AC, AD, AE |
| Colonized with the same ST throughout | 8 | F, G, M, Q, S, T, U, AF |
| Colonized with three STs, sequentially | 6 | E, J, K, O, W, AA |
| Colonized with the same ST but variable antimicrobial resistance gene profile | 2 | D, N |
| Complex colonization[a] | 5 | A, Z, I, R, L |

[a]Two patients (A, Z) colonized with two STs, one or both at a time; one patient (I) colonized with four STs, one at a time or mixed; one patient (R) colonized with three STs singly or mixed; and one patient (L) colonized with two STs. In this patient, at single points, colonization with different colony types of the same ST but different resistance types or different STs with different resistance types.

It was concluded that further clarity on the origin of diversity should be better obtained by whole-genome sequencing (WGS) (17). In the current study where WGS was used, the question of mutation giving rise to a new strain could not be established.

## Conclusions

This study has demonstrated rectal colonization of ICU patients with different patterns of colonization involving many STs of *A. baumannii*. However, the predominant ST2 colonized all patients at some time during hospitalization. Further studies, where multiple colonies are picked and their closed genome analyzed at sampling points, are required to elucidate the evolution of antimicrobial resistance and new strains.

## ACKNOWLEDGMENTS

This work received no specific funding.

D.B. performed bioinformatic analysis; G.P.C. and L.L. performed sequencing; G.A.-H. performed bacterial isolate collection; V.O.R. provided clinical supervision; and M.J.A. supervised the project. All authors read and approved the final version of the manuscript.

We have no conflicts of interest to declare.

## AUTHOR AFFILIATIONS

[1]Microbiological Diagnostic Unit Public Health Laboratory, Peter Doherty Institute for Infection and Immunity, The University of Melbourne, Melbourne, Victoria, Australia
[2]Melbourne Bioinformatics, The University of Melbourne, Carlton, Victoria, Australia
[3]Department of Microbiology and Immunology, Peter Doherty Institute for Infection and Immunity, The University of Melbourne, Melbourne, Victoria, Australia
[4]Department of Microbiology, College of Medicine, Kuwait University, Jabriya, Kuwait

## PRESENT ADDRESS

Vincent O. Rotimi, Center for Infection Control and Patient Safety, College of Medicine, University of Lagos, Idi-Araba, Lagos, Nigeria

## AUTHOR ORCIDs

M. John Albert  http://orcid.org/0000-0001-5794-7500

## AUTHOR CONTRIBUTIONS

Dieter Bulach, Formal analysis, Software, Writing – review and editing | Glen P. Carter, Methodology | Lucy Li, Methodology | Ghayda Al-Hashem, Investigation, Methodology, Resources | Vincent O. Rotimi, Conceptualization, Supervision | M. John Albert, Conceptualization, Project administration, Resources, Supervision, Writing – original draft

## DATA AVAILABILITY

The data supporting the conclusions of this article are within the article and its additional files. Sequence data are available in the GenBank repository at the National Center for Biotechnology Information (NCBI) under the BioProject accession number PRJNA791537.

## ETHICS APPROVAL

The study was approved by the ethics committee of the Ministry of Health, State of Kuwait. Informed consent was obtained from the caretakers of the patients.

## ADDITIONAL FILES

The following material is available online.

## Supplemental Material

**Fig. S1 (Spectrum02191-23-s0001.pdf).** Phylogenetic tree showing the relationship between the 202 ST2 *A. baumannii* isolates from this study and 288 ST2 *A. baumannii* isolates for which a closed genome sequence is available at NCBI.
**Table S1 (Spectrum02191-23-s0002.docx).** Scheme for isolation of *A. baumannii*.
**Table S2 (Spectrum02191-23-s0003.xlsx).** Sequencing information, genome assembly, and ST of the 269 *A. baumannii* isolates.
**Table S3 (Spectrum02191-23-s0004.xlsx).** List of *A. baumannii* ST2 closed genomes.
**Table S4 (Spectrum02191-23-s0005.docx).** Antimicrobial resistance genes.
**Table S5 (Spectrum02191-23-s0006.xlsx).** Correlation between antimicrobial resistance phenotype and antimicrobial resistance genotype.
**Table S6 (Spectrum02191-23-s0007.xlsx).** Sequence type and antimicrobial resistance gene profile.
**Table S7 (Spectrum02191-23-s0008.docx).** Colony morphology and genotype.

## Open Peer Review

**PEER REVIEW HISTORY (review-history.pdf).** An accounting of the reviewer comments and feedback.

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
