## [Reviewer comments · Microbiology Spectrum]

Microbiology Spectrum

The whole genome molecular epidemiology of sequential isolates of *Acinetobacter baumannii* colonizing the rectum of patients in an adult intensive care unit of a tertiary hospital.

Dieter Bulach, Glen Carter, Lucy Li, Ghayda Al Hashem, Vincent Rotimi, and M. John Albert

Corresponding Author(s): M. John Albert, Kuwait University Faculty of Medicine

Review Timeline:

Submission Date:	May 24, 2023
Editorial Decision:	August 3, 2023
Revision Received:	August 30, 2023
Accepted:	September 7, 2023

Editor: Varadharajan Sundaramurthy

Reviewer(s): The reviewers have opted to remain anonymous.

Transaction Report:

DOI: <https://doi.org/10.1128/spectrum.02191-23>

August 3, 2023

Prof. M. John Albert
Kuwait University Faculty of Medicine
Jabriya
Kuwait

Re: Spectrum02191-23 (Whole genome molecular epidemiology of sequential isolates of *Acinetobacter baumannii* colonizing the rectum of patients in an adult intensive care unit of a tertiary hospital.)

Dear Prof. M. John Albert:

Thank you for submitting your manuscript to Microbiology Spectrum. Your manuscript was reviewed by two experts in the field. Please find below their comments, which I think you should be able to address. It will be good to also get a sense of the sampling frequency, i.e. if some individuals were sampled more frequently than others. When submitting the revised version of your paper, please provide (1) point-by-point responses to the issues raised by the reviewers as file type "Response to Reviewers," not in your cover letter, and (2) a PDF file that indicates the changes from the original submission (by highlighting or underlining the changes) as file type "Marked Up Manuscript - For Review Only". Please use this link to submit your revised manuscript - we strongly recommend that you submit your paper within the next 60 days or reach out to me. Detailed instructions on submitting your revised paper are below.

Link Not Available

Sincerely,

Varadharajan Sundaramurthy

Journals Department
Reviewer comments:

Reviewer #1 (Comments for the Author):

Thank you for the opportunity to review - "Whole genome molecular epidemiology of sequential isolates of *Acinetobacter baumannii* colonizing the rectum of patients in an adult intensive care unit of a tertiary hospital". In this manuscript, the authors present genomic analysis of 269 sequential isolates of *A. baumannii* from 32 ICU patients at a tertiary care hospital. A majority of the isolates (202/269) belonged to a single strain type (ST2). All isolates carried antibiotic resistance genes (2-17) - particularly those conferring resistance to beta lactam and aminoglycoside antibiotics.

I have a few queries and suggestions on the presentation of the results

1. It appears (but is not completely clear) that the same cohort of patients and isolates have been used in previous studies : Al-Hashem G, Rotimi VO, Albert MJ. Genetic relatedness of serial rectal isolates of *Acinetobacter baumannii* in an adult intensive care unit of a tertiary hospital in Kuwait. *PLoS354 One*. 2020;15: e0230976.
Al-Hashem G, Rotimi VO, Albert MJ. Antimicrobial resistance of serial isolates of *Acinetobacter baumannii* colonizing the rectum of adult intensive care unit patients in a teaching hospital in Kuwait. *Microb Drug Resist*. 2021; 27:64-72

It would be good to understand how the present study builds up on the previous observations

2. Did the antimicrobial resistance genes in a particular background strain type increase with time? For patients in whom the no recolonization or mixed infection was suspected, it would be good to see whether sequential isolates had increased acquisition of AMR genes?

3. The methodology for phylogenetic analysis has not be described adequately. The methods for this needs to be expanded.

4. If the clinical cohort is the same as the previously described cohort (see point 1) then having the genotype to phenotype correlation for the antibiotics is essential and should be presented.

5.The sequences pf/reference for the gyB gene primer sets would be good to include

6. It would be good to have a figure showing the sampling timepoint - for example instead of denoting samples are the 5th sample it would be good to know that the sample was collected 20 days from the initial sampling time point.

7. Lines 271-275 of Discussion: Given that single colony sampling was predominantly used, this may have led to an underestimate of colonization with multiple STs.

Further investigation of simultaneous colonization would require the characterization of multiple isolates from each swab sample

In their previous work, the authors have suggested that the morphological features of the colony could suggest genotype differences - can this be used as a measure in the current study to understand the genetic diversity/heterogeneity that the authors suggest here?

8. The study was conducted in a single hospital, was there a spatial-epidemiological clustering of the strains in any way?

9. Is it possible that the presence of ST2 in majority of the isolates has more to do with the single site - I.e the presence and persistence of this strain in the hospital rather than the more general claim that this is best colonizer? Did all mixed infections become predominantly ST2 at the end of the study?

10. The phylogenetic trees could be presented better - probably by collapsing some branches, adding bootstrap support values (if trees are maximum likelihood trees), adding/overlaying metadata of country, time and resistance profiles on the tree itself.

11. Were all individuals sampled the same number of times? How much of the Strain distribution is impacted by differences in sampling?

12. The authors report that - "Within STs, there were often groups of isolates with the same antimicrobial resistance gene profile." - it would be good to see this summarized in a figure with cumulative frequencies for example.

13. "Table 1 shows the distribution of STs detected in each patient." - It would be good to also see this as a frequency distribution and also changes for a person over time

14. Discussion lines 202-203: The genome characteristics of our *A. baumannii* isolates were like those published previously (Park et al 2011; Liu et al 2014; Michiels et al 2016; Wibberg et al 2018).
Can the authors clarify what these features are?

15. Did the authors perform any analysis on mutations acquired by a strain in a person over time?

16. The methods have not been described in enough detail for the analyses presented to be replicated.

)

Reviewer #2 (Comments for the Author):

Bulach and colleagues performed a longitudinal genomic epidemiology investigation in the Intensive Care Unit (ICU) of the Mubarak Al Kabir Hospital (Kuwait). They included in their study the patients who showed a persistent colonization of *Acinetobacter baumannii*, performing longitudinal rectal swabs and WGS-based typing of the isolates. The most frequency Sequence Type (ST) they found was ST2 (not surprisingly), but other STs were also isolated. They also found patients colonized by multiple lineage of the bacterium. Correctly they stated at the end of the manuscript that this aspect is very interesting and that further investigations are necessary (e.g. using third generation sequencing approach).

The experimental design is sound and the manuscript well written. The results are interesting for the scientific community.

I can only suggest some minor changes in the text. In particular I suggest to capitalize the text indicated by the acronyms

e.g.

LINE 40: intensive care units have to be capitalized (the same in all occurrences in the manuscript)

LINE 45: Multilocus sequence typing should be "Multi-locus Sequence Typing"

LINE 46: sequence types should be capitalized

I also suggest replacing "development" with "selection" at LINE 87.

Staff Comments:

Preparing Revision Guidelines

Please return the manuscript within 60 days; if you cannot complete the modification within this time period, please contact me. If you do not wish to modify the manuscript and prefer to submit it to another journal, please notify me of your decision immediately so that the manuscript may be formally withdrawn from consideration by Microbiology Spectrum.

27/8/2023

Professor Varadharajan Sundaramurthy
Editor, Microbiology Spectrum

Re: The manuscript entitled "Whole genome molecular epidemiology of sequential isolates of *Acinetobacter baumannii* colonizing the rectum of patients in an adult intensive care unit of a tertiary hospital" (Spectrum02191-23)

Dear Dr. Sundaramurthy,

Thank you for forwarding the reviewers' comments. Our responses to the comments are addressed below. Please refer to the "Marked-Up" manuscript to view the changes.

Reviewer 1

Thank you for the opportunity to review - "Whole genome molecular epidemiology of sequential isolates of *Acinetobacter baumannii* colonizing the rectum of patients in an adult intensive care unit of a tertiary hospital". In this manuscript, the authors present genomic analysis of 269 sequential isolates of *A. baumannii* from 32 ICU patients at a tertiary care hospital. A majority of the isolates (202/269) belonged to a single strain type (ST2). All isolates carried antibiotic resistance genes (2-17) - particularly those conferring resistance to beta-lactam and aminoglycoside antibiotics.

Response

We thank the referee for this kind assessment.

I have a few queries and suggestions on the presentation of the results

1. It appears (but is not completely clear) that the same cohort of patients and isolates have been used in previous studies: Al-Hashem G, Rotimi VO, Albert MJ. Genetic relatedness of serial rectal isolates of *Acinetobacter baumannii* in an adult intensive care unit of a tertiary hospital in Kuwait. PLoS One. 2020;15: e0230976.

Al-Hashem G, Rotimi VO, Albert MJ. Antimicrobial resistance of serial isolates of *Acinetobacter baumannii* colonizing the rectum of adult intensive care unit patients in a teaching hospital in

Kuwait. Microb Drug Resist. 2021; 27:64-72

It would be good to understand how the present study builds up on the previous observations

Response

True (L 101-102).

To understand how the present study builds up on the previous observations, please see L 315-329.

2. Did the antimicrobial resistance genes in a particular background strain type increase with time? For patients in whom no recolonization or mixed infection was suspected, it would be good to see whether sequential isolates had increased acquisition of AMR genes?

Response

Please see L 213-217.

3. The methodology for phylogenetic analysis has not been described adequately. The methods for this need to be expanded.

Response

This has been done (L128-130, L498-502).

4. If the clinical cohort is the same as the previously described cohort (see point 1) then having the genotype to phenotype correlation for the antibiotics is essential and should be presented.

Response

This correlation is shown (L196-205 & L303-309).

5. The sequences pf/reference for the gyB gene primer sets would be good to include

Response

Included (L116).

6. It would be good to have a figure showing the sampling timepoint - for example instead of denoting samples as the 5th sample it would be good to know that the sample was collected 20 days from the initial sampling time point.

Response

This information was included in a previous publication in a table (Al-Hashem et al 2020, PLOS ONE). To avoid duplication, we have included a modified table (Table S1) with the reference.

7. Lines 271-275 of Discussion: Given that single colony sampling was predominantly used, this may have led to an underestimate of colonization with multiple STs. Further investigation of simultaneous colonization would require the characterization of multiple isolates from each swab sample

In their previous work, the authors have suggested that the morphological features of the

colony could suggest genotype differences - can this be used as a measure in the current study to understand the genetic diversity/heterogeneity that the authors suggest here?

Response

Please see L219-223 &L320-325.

8. The study was conducted in a single hospital, was there a spatial-epidemiological clustering of the strains in any way?

Response

This was not investigated.

9. Is it possible that the presence of ST2 in majority of the isolates has more to do with the single site - I.e the presence and persistence of this strain in the hospital rather than the more general claim that this is best colonizer? Did all mixed infections become predominantly ST2 at the end of the study?

Response

Please see L273-274.

Mixed infections did not become predominantly ST2 at the end of the study. Please see Table S1.

10. The phylogenetic trees could be presented better - probably by collapsing some branches, adding bootstrap support values (if trees are maximum likelihood trees), adding/overlying metadata of country, time, and resistance profiles on the tree itself.

Response

Additional relevant information has been added to the taxon label for both Fig 1 and Fig S1.

11. Were all individuals sampled the same number of times? How much of the Strain distribution is impacted by differences in sampling?

Response

Please see L111 &L317-318.

12. The authors report that - "Within STs, there were often groups of isolates with the same antimicrobial resistance gene profile." - it would be good to see this summarized in a figure with cumulative frequencies for example.

Response

Please see L213-216.

13. "Table 1 shows the distribution of STs detected in each patient." - It would be good to also see this as a frequency distribution and also changes for a person over time

Response

Please see L189-192.

14. Discussion lines 202-203: The genome characteristics of our *A. baumannii* isolates were like those published previously (Park et al 2011; Liu et al 2014; Michiels et al 2016; Wibberg et al 2018).

Can the authors clarify what these features are?

Response

Please see L240-242.

15. Did the authors perform any analysis on mutations acquired by a strain in a person over time?

Response

Please see L303-307.

16. The methods have not been described in enough detail for the analyses presented to be replicated.

Response

This has been already dealt with.

Reviewer #2 (Comments for the Author):

Bulach and colleagues performed a longitudinal genomic epidemiology investigation in the Intensive Care Unit (ICU) of the Mubarak Al Kabir Hospital (Kuwait). They included in their study the patients who showed a persistent colonization of *Acinetobacter baumannii*, performing longitudinal rectal swabs and WGS-based typing of the isolates. The most frequency Sequence Type (ST) they found was ST2 (not surprisingly), but other STs were also isolated. They also found patients colonized by multiple lineage of the bacterium. Correctly they stated at the end of the manuscript that this aspect is very interesting and that further investigations are necessary (e.g. using third generation sequencing approach).

The experimental design is sound and the manuscript well written. The results are interesting for the scientific community.

Response

We thank the referee for this kind assessment.

I can only suggest some minor changes in the text. In particular I suggest to capitalize the text indicated by the acronyms

e.g.

LINE 40: intensive care units have to be capitalized (the same in all occurrences in the manuscript)

Response

Please see L2, 40, 61, 76.

LINE 45: Multilocus sequence typing should be "Multi-locus Sequence Typing"

Response

Please see L45.

LINE 46: sequence types should be capitalized

Response

Please see L46.

I also suggest replacing "development" with "selection" at LINE 87.

Response

Please see L88.

We hope that we have satisfactorily answered all the comments.

With best regards

Professor M. John Albert
Department of Microbiology
College of Medicine
Kuwait University
Kuwait

September 7, 2023

Prof. M. John Albert
Kuwait University Faculty of Medicine
Jabriya
Kuwait

Re: Spectrum02191-23R1 (The whole genome molecular epidemiology of sequential isolates of *Acinetobacter baumannii* colonizing the rectum of patients in an adult intensive care unit of a tertiary hospital.)

Dear Prof. M. John Albert:

Your manuscript has been accepted, and I am forwarding it to the ASM Journals Department for publication. You will be notified when your proofs are ready to be viewed.

Sincerely,

Varadharajan Sundaramurthy
Editor, Microbiology Spectrum
